# Low Vitamin D States Observed in U.S. Marines and Navy Sailors with Early Multi-Symptom Illness

**DOI:** 10.3390/biom10071032

**Published:** 2020-07-11

**Authors:** Sean R. Maloney, Paula Goolkasian

**Affiliations:** 1Navy Mobilization Processing Site, Deployment Processing Command-East, W.G. (Bill) Hefner VA Medical Center, Salisbury, NC 28144, USA; 2Department of Psychological Science, University of North Carolina at Charlotte, Charlotte, NC 28223, USA

**Keywords:** immune modulation, vitamin D deficiency, multi-symptom illness, Epstein-Barr virus, sensory neuritis, deployment history, vestibular dysfunction

## Abstract

Research has implicated immune system inflammation as an underlying etiology of multi-symptom illnesses, and vitamin D has been shown to have a significant role in immune system function. In this retrospective review performed on the medical charts of service members who presented with signs and symptoms of multi-symptom illnesses, we focused on serum 25(OH)D_3_ levels and looked for associations of vitamin D status (deficient, insufficient, and normal) with age (20–31 years versus 31–56 years) and deployment status (war zones versus other). Two groups (U.S. Marines and Navy Sailors) were sampled and both showed high incidences of below normal vitamin D levels. However, with the Marines, age-related differences in serum levels (*p* = 0.009) were found only among those who deployed to Iraq/Afghanistan in comparison to those in non-combat locations. The comparison within the Navy sample showed that mobilized sailors had lower 25(OH)D_3_ levels than the group that did not deploy (*p* = 0.04). In addition, 100% of the sailors who deployed had below normal levels versus only 33% in the cadre group. The data suggest that personnel returning from a war zone with signs of early multi-symptom illness should be checked for low vitamin D status.

## 1. Introduction

Many U.S. service members returning from the wars involving Iraq and Afghanistan since 1990–1991 have returned with chronic multi-symptom illnesses [1] (often disabling) for which the pathophysiology or etiology is unknown. Recent research includes the finding that some multi-symptom illnesses such as Gulf War illness (GWI) and myalgic encephalomyelitis or chronic fatigue syndrome (ME/CFS) are associated with pre-lytic herpes virus partial reactivation [2,3,4,5]. This partial reactivation involves the synthesis of herpetic virus dUTPase protein antigens with resulting stimulation of inflammatory intermediates. There has been independent documentation of increased inflammatory cytokines in veterans with GWI [6,7,8]. There have also been clinical studies which suggest an atypical or partial reactivation of herpetic viruses as an etiology of chronic multi-symptom illnesses including ME/CFS and in those with neuropathic chronic pain syndromes. Non routine, experimental, antiviral therapy has been used with moderate success to treat patients with these chronic multi-symptom illnesses [9,10,11,12].

The active form of vitamin D, 1,25(OH)_2_ D_3_ is a hormone that can be produced outside the kidneys in immune system cells, and this hormone uses vitamin D receptors on activated lymphocytes, macrophages, and dendritic cells [13,14,15,16]. Persistent low serum levels of active vitamin D precursor, 25(OH)D_3_ may chronically impair the immune system in individuals with multi-symptom illnesses in two ways. First, in low vitamin D states, the immune system may be unable to completely suppress latent viruses and other pathological microorganisms in part due to inadequate production of cathelicidin [17,18,19]. The administration of oral vitamin D_3_, cholecalciferol, has been shown to induce cathelicidin [20]. Second, vitamin D is a hormone that acts directly on the immune system, including B and T lymphocyte cells, to down regulate the inflammatory reaction triggered by latent viruses, including their antigens, or other microorganisms [13,14,21]. The presence of frequent and chronic low vitamin D states in U.S. veterans with multi-symptom illnesses treated at the Department of Veterans Affairs medical facilities has been previously reported [22,23].

The herpes Epstein–Barr virus has been studied with respect to pre-lytic herpetic dUTPase antigen production in patients with chronic multi-symptom illnesses [2,5]. An inverse relationship of vitamin D status with Epstein–Barr infection and autoimmunity has also been previously described [24]. There is growing evidence that low vitamin D status may increase the risk of acute mononucleosis in U.S. service members and others [25,26,27]. Lupus, an autoimmune disease, has been associated with increased dUTPase antigen levels in lupus nephritis [28]. Lupus has also been associated with Epstein–Barr infections and acute mononucleosis [29]. One female service member in our Camp Lejeune study group progressed from fatigue and a sensory neuritis in a limb associated with low vitamin D status to acute mononucleosis [26] and then to acute lupus with serious cardiovascular involvement.

A low vitamin D state may also decrease the immune system’s ability to produce the activated CD8+ T lymphocytes that attack Epstein–Barr virus infected B lymphocytes. A low vitamin D state may also increase the inflammatory response by CD8+ T lymphocytes in the presence of Epstein–Barr virus infected cells [24]. In young individuals (less than 26 years of age), serum 25 OH vitamin D_3_ levels correlated inversely to antibody reactivity against Epstein–Barr nuclear antigen [30]. Finally, vitamin D supplementation in persons wintering over in Antarctica has been shown to mitigate Epstein–Barr virus reactivation [31].

Idiopathic multi-symptom illnesses are typified by clusters of medically unexplained symptoms among which are symptoms of persistent musculoskeletal and neuropathic pain and severe fatigue [1,32,33,34]. Previous research conducted in the VA Health Care System has shown some success with relieving the symptoms of multi-symptom illness by correcting vitamin D deficiency in symptomatic patients [35]; and based on this research, vitamin levels were collected in this study. This paper presents a retrospective case series report of U.S. Marines and Navy Sailors with early multi-symptom illnesses who were processed through or who were assigned as cadre to the Navy Medical Processing Site (NMPS), U.S Marine Deployment Processing Command East (DPC-E), Camp Lejeune, NC. The purpose of the chart review was to document the number of symptomatic service members returning from deployments who had low normal to deficient serum 25(OH)D_3_ levels, and that purpose was later expanded to look at where they were deployed and age, because many of the younger U.S. Marines had vitamin D deficiency.

## 2. Materials and Methods

### 2.1. Participants

The charts from 105 U.S. Marines and 12 Navy Sailors who were either processed through, or were assigned as cadre to the NMPS, DPC-E, Camp Lejeune, N.C. between 2 September 2010 and 21 July 2011 were reviewed. Fourteen female Marines and 3 female Navy Sailors were included. All of the symptomatic U.S. service members in this study had serum 25(OH)D_3_ levels drawn at either the time of their medical screenings during demobilization or in the case of the DPC-E cadre during sick call medical evaluations. Common symptoms were persistent idiopathic musculoskeletal or neuropathic pain, decreased endurance, tinnitus, and fatigue. The most common signs found on the physical exam included joint tenderness, skin hyperalgesia to pinprick (sensory neuritis), and decreased balance upon heel to toe waking especially with the eyes closed [35,36]. The service members (with two NMPS Navy cadre exceptions) included in this chart review were evaluated and treated by one reserve navy medical officer who was mobilized to the medical unit of the Navy Mobilization Processing Site, DPC-E in August of 2010.

Comparisons of serum 25(OH)D_3_ levels were made in the US Marine service member group based on age and deployment history (deployed to the war zones of Iraq/Afghanistan or to other noncombat locations). Comparisons of serum 25(OH)D_3_ levels were made in the US Navy sailors based on whether they were assigned to the DPC-E as cadre (no deployment) or processed through the DPC-E as non-cadre (deployed). The non-cadre U.S. Navy sailors in this study were typically reserve U.S. Navy corpsman recalled to active duty and assigned to deploying combat U.S. Marine Corps units or were reserve U.S. Navy Seabees recalled to active duty for service in Iraq or Afghanistan.

All serum 25(OH)D_3_ levels included in this study were initial serum 25(OH)D_3_ levels and no service members in this study had a known prior history of low vitamin D status. This chart review included all the initial serum 25(OH)D_3_ levels ordered on DPC-E personnel during the study period by the medical officer noted above. None of 117 service members, whose charts were reviewed, were on vitamin D supplementation or on a medication that interfered with vitamin D_3_ metabolism [37]. The Human Use Review Committee and the Research Committee at the Naval Medical Center, Portsmouth, Virginia approved this study. Patient records were de-identified prior to data analysis. (Study: NHCL.2012.0007—“Retrospective Record Review of Serum 25(OH) vitamin D_3_ Levels in US Marines and Navy Sailors”—Completed 4 June 2012).

A total of 439 Marines, 131 Navy Sailors, and a large number of civilian contractors (not included in this study) were processed through the Navy Mobilization Processing Site (NMPS), U.S. Marine Deployment Processing Command-East during the study period. There were approximately 44 U.S. Marine cadre and 14 U.S. Navy cadre assigned to the DPC-East during the study period. Among the U.S. Navy cadre there were three medical providers assigned to the NMPS, DPC-E during this period to evaluate personnel.

### 2.2. Vitamin D Levels

Serum 25(OH)D_3_ levels were converted to seasonally adjusted, annual mean, serum 25(OH)D_3_ levels using the “25 Hydroxyvitamin D Calculator for Seasonal Adjustment” developed at the Kidney Research Institute, University of Washington [38,39]. Serum specimens were evaluated by outside reference labs (Lab Corp of America, Burlington, NC, USA—2 September 2010 to 26 January 2011 and Quest Lab, Jersey City, NC, USA—27 January 2011 to 21 July 2011). Of the 117 samples drawn from U.S. Marines and Navy Sailors during this study, 65 were processed by LabCorp and 52 were processed by Quest Lab.

To categorize the serum data into normal, insufficient, and deficient levels, the reference ranges established by both reference labs on the seasonally adjusted mean serum 25(OH)D_3_ data were used. They were as follows: vitamin D deficiency (<20 ng/mL), vitamin D insufficiency (20 ng/mL to <30 ng/mL), and normal (30 ng/mL to 100 ng/mL). The guideline for vitamin D deficiency recommended by the Endocrine Society is in contrast to the guideline for deficiency of 30 nm/L or 12.02 ng/mL recommended by the Institute of Medicine. The serum specimens were analyzed for serum 25(OH)D_3_ at Lab Corp of America by immunochemiluminometric assay performed on the DiaSorin LIASONR instrument and at Quest Diagnostic Nichols Institute Lab by liquid chromatography/tandem mass spectroscopy (LC/MS/MS).

### 2.3. Data Analysis

Because of differences between Navy and Marine physical standards, training regimens, and known deployment history, the Navy and Marine data were treated as separate groups when looking for relationships with serum 25(OH)D_3_ levels. There were 47 Marines in the younger group and 58 in the older group. The two age groups (20–31 years and 32–56 years) were chosen for comparison of the U.S. Marine data to find the optimal age groups to reflect differences in 25(OH)D3 levels between younger and older service members while balancing the number of participants in each of the deployment conditions. Age groups were further divided based on those who deployed or had a history of a post 9 November 2001 deployment to the war zones of Iraq or Afghanistan, and those deployed to other locations or who had no post 9 November 2001 war zone deployment. Examples of other deployment locations might include Bahrain, United Arab Emirates, Kuwait, Djibouti, or USCENTCOM, MacDill Air Force Base, Tampa, Florida, etc. With both age groups, however, there were many more deployed to a war zone than to other locations. For the younger group, there were 31 in the war zone group and 16 in the other group; and in the older group, 44 were in the war zone group, while 14 were in the other group.

Because of the imbalance in the numbers that were deployed in the two categories, our analysis for the Marine data focused on comparing age groups within each of the deployment categories. The analyses used a general linear model to covary the gender of the service member and the lab that was used to analyze the serum to look for age group differences in seasonally adjusted, annual mean, serum 25(OH)D_3_ among those deployed to a war zone and those deployed to other locations. In addition, Chi-square tests were used to test if the distribution across the three vitamin D categories varied by age and deployment status.

The U.S. Navy Sailors, because of their small group size, were divided based only on their deployment status. They were evenly divided (6 and 6) between those who deployed and those who did not deploy. The deployed U.S. Navy sailors in this study were typically reserve medical corpsman assigned to combat Marine units or U.S. Navy Seabees. A Mann–Whitney U test was used to compare the group differences in seasonally adjusted vitamin D levels, and a Chi-square test looked for group differences across the three vitamin D level categories. A significance level of *p* equal to or less than 0.05 was used for all statistical tests. Statistical analyses were performed using SPSS Statistics Version 23 (IBM Corporation, Armonk, NY, USA). Participant characteristics within each of the groups are presented in Table A1.

## 3. Results

### 3.1. Marine Data

Among the 105 Marines, 27% of the 25(OH)D_3_ levels were categorized as deficient (<20 ng/mL) and 36% insufficient (20 ng/mL to <30 ng/mL). Box plots showing the distribution of seasonally adjusted, annual mean, serum 25(OH)D_3_ ng/mL levels for the age groups within each of the deployment categories are presented in Figure 1, while Table 1 provides some additional statistics on each of the groups. Among those who were deployed or had a history of a post 9 November 2001 deployment to the war zones of Iraq or Afghanistan, there was a significant difference between the younger and older Marines in serum 25(OH)D_3_ levels, *F* = 7.42, *p* = 0.008. The younger Marines had lower serum 25(OH)D_3_ levels than the older group on average, and a higher percentage of those categorized as deficient (42%) and insufficient (32%). By comparison, the older group had 18% categorized as deficient and 36% insufficient. However, an age group related difference in the distribution of vitamin levels was not found to be quite large enough for significance when the data from the two deployment categories (war zone and other) were combined for each age group, *Chi Square*(2, *n* = 75) = 5.63, *p* = 0.06.

Among those Marines who were deployed to locations outside of war zones, there were no age-related differences in seasonally adjusted mean serum vitamin D levels, *F* = 0.65, *p* = 0.801, and there were no differences between the younger and older Marines in the distribution of their data across the vitamin D categories, *Chi Square*(2, *n* = 30) = 0.10, *p* = 0.95. For the younger age group, 25% had deficient vitamin D levels and 38% were insufficient, while the older group had 21% deficient and 43% insufficient. Table 2 sums across the age groups and shows how each of the deployment categories were distributed across the vitamin D categories. Overall (combining all ages), there were no differences in the vitamin D distribution between those deployed to a war zone and those deployed to other locations, *Chi Square*(2, *n* = 105) = 0.35, *p* = 0.84.

Data in the box plots are for the two age groups who either deployed or had a history of deployment to a war zone (WZ) or who deployed to another location or who had no history of deployment to Iraq or Afghanistan (other).

### 3.2. Navy Data

The data from the 12 Navy Sailors showed that 50% were categorized as deficient and 17% insufficient. Seasonally adjusted 25(OH)D_3_ levels (ng/mL) taken from the Navy Sailors who were deployed compared to those who did not deploy are in Figure 2. Results of the Mann–Whitney U test (*p* = 0.04) showed differences. The group of Navy Sailors who deployed had lower 25(OH)D_3_ levels (*Mdn* = 18, *IQR* = 5.75) than the group that did not deploy (*Mdn* =31.5, *IQR* = 22). When the seasonally adjusted vitamin D data were categorized into levels, every one of the Sailors who deployed had below normal levels. There was an 83% deficiency rate and a 17% insufficient rate. By comparison, for those assigned to the NMPS, DPC-E as cadre, only 33% were below normal. Figure 3 shows the significant group differences in the distribution of the data across the vitamin D categories, *Chi Square* (2, *n* = 12) = 6.67, *p* = 0.036. A Mann–Whitney U test confirmed that these two groups of Navy Sailors were not found to differ in age, *p* = 0.59. Median ages for those who deployed verses those who did not deploy are as follows: 31.5 years (*IQR* = 12) and 34.5 years (*IQR* = 17).

## 4. Discussion

Prior to September 2010, serum 25(OH)D_3_ levels had been rarely if at all checked in symptomatic U.S. Marines and Navy Sailors processed through the NMPS at Camp Lejeune, NC. Among the 105 symptomatic U.S. Marines who had levels checked, 27% of their seasonally adjusted annual mean serum 25 OH vitamin D_3_ levels were in the deficient category, requiring treatment. Similarly, 50% of U.S. Navy Sailors who had serum 25 OH vitamin D_3_ levels were in the deficient category, and these Navy Sailors required initiation of treatment. This case study raises but does not answer the question, “Does mobilization and deployment to a war zone or any other location increase the risk of vitamin D deficiency?” A prospective controlled study with a control group made up of asymptomatic U.S Marines or Navy Sailors who deployed in a similar manner is needed to answer the question.

However, there is some evidence that a recent history of deployment of service members to a war zone may be associated with lower serum 25(OH)D_3_ levels than those without a known recent deployment. A study of 684 active duty U.S. Army soldiers at Ft. Bragg, NC and U.S. veterans living in the area of Ft. Bragg underwent vitamin D testing at the Womack Army Medical Center and were noted to have a rate of vitamin D deficiency of 17% (mean 25(OH)D_3_ level of 29.2 ng/mL) [40]. In contrast, a study of serum 25(OH)D_3_ levels in deployed U.S. service members in two matched cohorts (*n* = 495 each) using serum samples from the Defense Medical Surveillance System (DMSS) drawn near the time of deployment had rates of vitamin D deficiency of 33.5% (mean 25(OH)D_3_ level of 24.5 ng/mL) and 31.3% (mean 25(OH)D_3_ level of 24.8 ng/mL) [41]. Additionally, a more recent study of British soldiers, mobilized to Afghanistan to maintain stability after the height of the conflict, found serum 25(OH)D_3_ levels increased and percent body fat decreased significantly with bright summer sunshine and vigorous activity. Their serum 25(OH)D_3_ vitamin D level response to the environment of Afghanistan and vigorous activity was normal and contrary to the findings of our study in symptomatic service members returning from Iraq and Afghanistan near the height of the conflict [42]. 

The reason or reasons why younger U.S. Marines who deployed to a combat zone had such a high rate of vitamin D deficiency (42%) and significantly lower seasonally adjusted annual mean serum 25(OH)D_3_ levels than older U.S. Marines who deployed to a combat zone is not known. The deployments of older U.S. Marine cadre may have occurred further in the past and thus had a smaller effect. In addition, the experiences including stress levels of the older, higher ranked U.S. Marines during the deployment may have been different.

A previous nutritional study, which included evaluation of serum 25(OH)D_3_ levels, was performed on young U.S. Soldiers during a 21-day U.S. Army Special Forces Assessment and Selection program [43]. The extreme physical, nutritional, and psychological stressors of the Special Forces Assessment and Selection Program are designed to evaluate these U.S. soldiers’ ability to learn the skills they will need to use during very stressful deployments in war zones and to see how they might function in a very stressful environment. The Special Forces Assessment and Selection study showed a consistent drop in serum 25(OH)D_3_ levels. Throughout the entire 21-day course, mean serum levels (*n* = 37) dropped by one standard deviation.

This case study also raises other questions, namely “Are symptomatic mobilizing and demobilizing U.S service members at higher risk of vitamin D deficiency and of requiring long term care for chronic multi-symptom illnesses through the VA Health Care System than asymptomatic mobilizing and demobilizing service members?” The service members in this study were selected to have serum 25(OH)D_3_ levels based on their signs and symptoms of early multi-symptom illnesses. The sign of hyperalgesia to pin prick (sensory neuritis) is attributed to inflammation of the sensory ganglia from zoster dUTPase antigens based on the pre-lytic viral immune inflammatory response model [2,12,28,35]. The sign of decreased balance and symptom of tinnitus is similarly based on presumed inflammation of the middle ear by non-suppressed herpes simplex 1 virus dUTPase antigens [36]. Finally, does younger age increase the risk of lower vitamin D status in these U.S. service members?

A larger prospective study is needed to track symptomatic and asymptomatic U.S. service members through the mobilization and later demobilization process, where these service members act as their own controls and where seasonally adjusted annual mean serum 25(OH)D_3_ levels, herpetic virus dUTPase levels, and immune system inflammatory cytokine levels to these antigens are measured repeatedly. In addition, to match a particular herpetic virus with the location and intensity of inflammation consistent with service members signs and symptoms and to measure a response to treatment (i.e., for example correction of inadequate vitamin D status for immune system health), new methods of synthesizing, tagging, and imaging specific antibodies to these herpetic antigens in the body are also necessary.

In this sample of service members, both age and deployment status appear to work together to lower serum 25(OH)D_3_ levels in U.S. Marines assigned to or processed through the Deployment Processing Command-East at Camp Lejeune, NC. When comparing the groups of Marines by age, keeping deployment status uniform, younger Marines who deployed to Afghanistan or Iraq had lower post deployment serum 25(OH)D_3_ levels than older Marines with a similar deployment history. Those U.S. Marines who deployed to other noncombat locations did not show any age-related difference in serum 25(OH) vitamin D_3_ levels.

This retrospective chart review study is limited by a relatively small number of cases, and the lack of normal control study subjects. Another limitation was the small number of female subjects and the lack of data on estrogen supplementation which can increase serum 25(OH)D_3_ levels. In addition, ethnicity and level of skin pigmentation information was not available at the time of the chart review and was not controlled. Ethnicity and level of skin pigmentation could affect the level of serum 25(OH)D_3_ levels recorded in the study because darker skin pigmentation is often associated with lower serum 25(OH)D_3_ levels. It is possible that more members who were deployed (and had lower vitamin D levels) were of non-European ancestry, which could have confounded the results. However, data from the defense department noted below show low levels of minority enrollment in the U.S. Marine Corps.

The U.S. Department of Defense Office of Diversity Management and Equal Opportunity [44] listed the percent of African American Officers in the U.S. Marine Corps as 5.0% in 2010 and 4.9% in 2011. African American enlisted personnel made up 10.9% of the enlisted population of the U.S. Marine Corps in 2010 and in 2011. The percent of U.S. Marine Hispanic officers was 6.6% in 2010 and 6.7% in 2011. The percent of U.S. Marine Hispanic enlisted personnel was 13.7% in 2010 and 14.8% in 2011. It is possible that there may have been more African American and Hispanic U.S. Marines in the younger group than the older group but this would not account for the discrepancy in low vitamin D status between the younger group that deployed to a war zone verses to a non-war zone.

The Department of Defense, Office of Diversity Management and Equal Opportunity [45] listed the percent of African American Officers in the U.S. Navy as 7.8% in 2010 and 7.7% in 2011. African American enlisted personnel made up 20.4% of the enlisted population of the U.S. Navy in 2010 and 19.9% in 2011. The percent of U.S. Navy Hispanic officers in 2010 was 6.2% in 2010 and 6.4% in 2011. The percent of U.S. Navy Hispanic enlisted personnel was 16.8% in 2010 and 17.7% in 2011. Overall, the ethnic representation of African Americans and Hispanics in the U.S. Marine Corps was lower than in the civilian population based on the 2010 U.S. census (African American population, 13.6% [46] and Hispanic population 16.3% [47]). In contrast the representation of enlisted African Americans in the U.S. Navy was considerably higher than in the U.S. population, and the representation of U.S. Navy enlisted Hispanics was equal to the civilian population. This relatively higher percent of these two minorities in the Navy could have influenced the differences between the serum 25(OH)D_3_ levels in the two groups studied.

The normal, insufficiency, and deficiency ranges used by our reference lab, the Endocrine Society, and this paper are higher than those recommended by the Institute of Medicine. One problem with trying to establish a true deficiency level for vitamin D is that it is not a vitamin but rather a hormone, and like other hormones the need for the hormone vitamin D fluctuates depending on an individual’s situation. A pregnant woman or an ICU patient with multiple fractures from trauma will have a greater need for active vitamin D than these same individuals without these conditions.

There is also significant seasonal variation in vitamin D precursor serum levels in most individuals, which means that if someone is on the border of insufficiency or deficiency in late August or early September (the peak in the northern hemisphere), then they will cross over into the deficiency range by late February or early March (the trough level). Finally, and perhaps most importantly, our current way of determining vitamin D adequacy depends on the measurement of vitamin D precursor levels in the serum but not on an individual’s need for the active vitamin D hormone. Thus, there is probably no single deficiency level at which true deficiency can be defined using serum 25(OH)D_3_ levels. For these reasons, using a higher cutoff level for vitamin D precursor to define deficiency makes more clinical sense until we have a better way to measure the fluctuating need for vitamin D and active vitamin D synthesis.

The accuracy of the vitamin D levels measured in this paper are strengthened by the fact that they were seasonally controlled by using calculated annual mean levels. Verification that service members, especially younger service members, deploying to a stressful war zone may be at higher risk for vitamin D deficiency will require larger prospective controlled studies, where before and after vitamin D levels are drawn together with additional measures taken to mitigate the many different personal, environmental, and nutritional factors known to influence vitamin D status.

## 5. Conclusions

This retrospective chart review documented frequent low and low normal levels of seasonally adjusted individual annual mean serum 25(OH)D3 levels in U.S. Marines and Navy Sailors with symptoms of early multi-symptom illnesses who were assigned to as cadre or processing through the U.S. Marine Deployment Processing Command-East, Camp Lejeune, N.C. between 2 September 2010 and 21 July 2011. The signs and symptoms of early multi-symptom illness in these U.S. Marines and Navy Sailors were consistent with a previously discovered increased immune inflammatory response from herpetic viral dUTPase antigens in others suffering from the chronic multi-symptom illnesses of Gulf War illness and chronic fatigue syndrome. 

Younger Marines who deployed to Iraq or Afghanistan had a lower mean serum 25(OH)D_3_ levels than older Marines who deployed to Iraq or Afghanistan. This difference in serum 25(OH)D_3_ levels was not seen between younger and older U.S. Marines who deployed to another location or who had no post 9 November 2001 deployment to Iraq or Afghanistan. Mobilized/deployed non-cadre U.S. Navy Sailors had very high rates of vitamin D deficiency and lower serum 25(OH)D_3_ levels than sailors assigned to the DPC-E as cadre. These findings suggest that serum 25(OH)D_3_ levels should be checked and treated in symptomatic U.S. Marines and Navy Sailors deploying to and returning from areas of conflict.

## Figures and Tables

**Figure 1 biomolecules-10-01032-f001:**
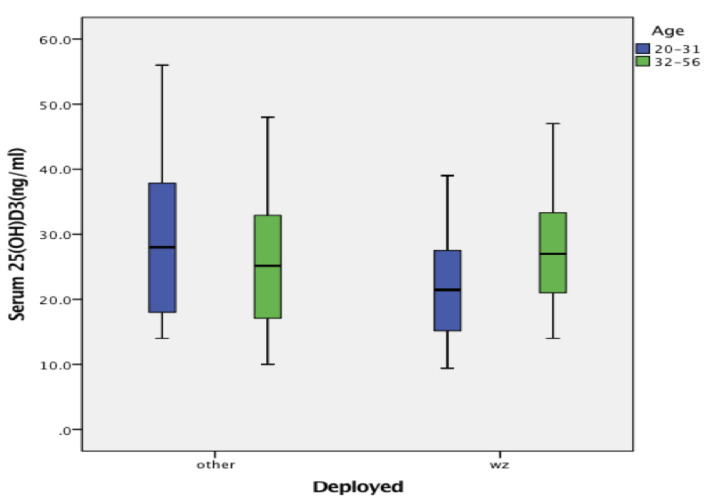
Seasonally adjusted serum 25(OH)D_3_ (ng/mL) levels drawn from the U.S. Marines.

**Figure 2 biomolecules-10-01032-f002:**
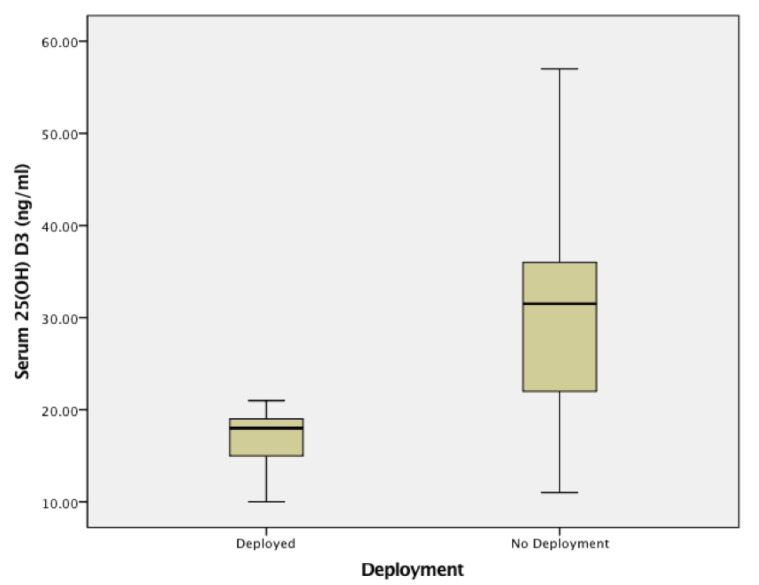
Seasonally adjusted serum 25(OH)D_3_ (ng/mL) levels drawn from the U.S. Navy Sailors. Data in the box plots compare serum 25(OH)D_3_ levels drawn from non-cadre Navy Sailors (*n* = 6) who processed through the Deployment Processing Command East (DPC-E), (typically demobilizing) from U.S. Marine units with those U.S. Navy Sailors (*n* = 6) who were not deployed but who were assigned to the DPC-E as cadre.

**Figure 3 biomolecules-10-01032-f003:**
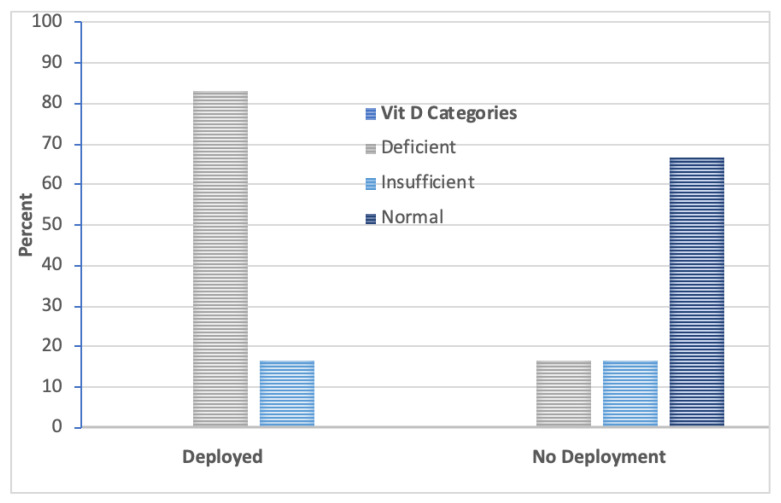
Percent of U.S. Sailors in each of the vitamin D Categories. There was a significant difference (*p* = 0.036) in the distribution of vitamin D levels between the deployment groups.

**Table 1 biomolecules-10-01032-t001:** Mean seasonally adjusted serum 25(OH)D_3_ ng/mL levels drawn from groups of U.S. Marines.

Deployed	Age	Mean	SD	*n*	95% CI LL	95% CI UL	*p*
Other	20–31	28.62	11.73	16	23.96	33.40	0.801
	32–56	27.57	10.52	14	21.58	32.56	
	Total	28.13	11.00	30	24.68	31.52	
Wz	20–31	22.84	8.42	31	19.48	26.19	0.008
	32–56	28.25	8.80	44	25.43	31.07	
	Total	26.01	9.00	75	23.35	27.74	
Total	20–31	24.81	9.94	47	22.86	28.61	
	32–56	28.08	9.15	58	25.04	30.78	
				105			

Other = Marines deployed to locations other than a war zone were not found to have age-related difference in serum levels, *F* = 0.65, *p* = 0.801. Wz = Marines deployed to the war zones of Iraq or Afghanistan were found to have an age-related difference in serum levels, *F* = 7.42, *p* = 0.008.

**Table 2 biomolecules-10-01032-t002:** Distribution of U.S. Marines across vitamin D categories. No difference in distribution by deployment groups (*Chi Square*(2, *n* = 105) = 0.35, *p* = 0.80).

Deployed		Vitamin D Category			Total
		Deficient	Insufficient	Normal	
Other	Count	7	12	11	30
	Percent	23%	40%	36%	100%
War Zone	Count	21	26	28	75
	Percent	28%	35%	37%	100%
Total	Count	28	38	39	105
	Percent	27%	36%	37%	100%

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
