# Peer review of "Low Vitamin D States Observed in U.S. Marines and Navy Sailors with Early Multi-Symptom Illness"

_biomolecules, 2020, doi:10.3390/biom10071032_

Round 1
Reviewer 1 Report
The present manuscript describes a retrospective case report of the vitamin D status of military personnel presenting with symptoms of multi-symptom illness at the medical facilities of Camp Lejeune, NC, USA. The authors sought to characterize levels of vitamin D adequacy in these patients and to examine whether history of deployment and age contributed to vitamin status.
While this is a worthwhile question, the present manuscript suffers from a number of methodological shortcomings that preclude making any conclusions about the observations reported by the authors. Perhaps most damaging is the lack of accounting for participant ethnicity and sex. Given that vitamin D is produced cutaneously upon exposure to sunlight, considering the participants’ skin colour and determining whether ethnic breakdown differed between younger and older participants, and those who were deployed to a war zone versus a non-conflict location, is crucial to better understand whether any real differences exist. Equally important, sex is associated with vitamin D status, and common medications that many women of reproductive age take (i.e. estrogen-containing hormonal contraceptives) are well documented to dramatically increase 25(OH)D plasma concentrations. The wide value spread in vitamin D concentrations documented among the navy sailors (Figure 2) makes one wonder, again, not just about the ethnic breakdown of those participants but also whether some of them were women taking hormonal contraceptives or other medications containing synthetic estrogen.
Aside from the points outlined above, there are also some other considerations that merit attention, as outlined below.
- Line 44: “Vitamin D3 is a hormone which can be produced outside the kidneys in immune system cells.” This statement is misleading. Vitamin D3 is cholecalciferol, which the kidneys and immune cells do not produce. The form of vitamin D that the authors are referring to is 1,25(OH)D, which is the biologically active form of vitamin D and results from the conversion of chole- or ergocalciferol (D2) into 25(OH)D, the accepted marker of vitamin D status, and subsequent hydroxylation into 1,25(OH)D.
- In that same paragraph, the authors need to clarify that “persistent low vitamin D states” are a low status of 25(OH)D, not 1,25(OH)D – which is tightly controlled and circulates at low levels in plasma. Perhaps a short paragraph on vitamin D metabolism would be worthwhile.
- Line 49: Cathelidicin is misspelled, and the authors imply that low cathelidicin production results from a low vitamin D status. Is there actual evidence that cathelidicin affects vitamin D production? If so, it should be cited directly.
- Section 2.2: Vitamin D Levels. The authors classify the participants as being deficient if they have blood 25(OH)D below 20 ng/ml, which is equivalent to 50 nmol/L. In fact, the Institute of Medicine of the National Academies defines actual deficiency as 25(OH)D below 30 nmol/L. While one could counter that there are differences of opinion across organizations on what constitutes a deficient vitamin D status, the authors also classify a normal status as the equivalent of 75 nmol/L to 250 nmol/L. Most vitamin D experts would consider 75 nmol/L to be optimal, rather than normal, and 250 nmol/L to be exceedingly high and potentially dangerous.
- Section 2.2: Vitamin D Levels. Vitamin D measurements were performed in two different labs, using an immunochemiluminometric assay in one case and liquid chromatography/tandem mass spectroscopy in the other. The authors need to indicate the degree of agreement between the two methods, and they should consider running statistical analyses that include method of measurement as a covariate. Along similar lines, it needs to be made clear how seasonal adjustments were calculated to generate a mean annual concentration of vitamin D. Alternately, the authors could consider not using the seasonally adjusted values and rather including season of blood draw as a covariate in their statistical analyses.
- Section 2.3: Data Analysis. It is not clear why those age groups were chosen. Why not split the participants by the median age, which would yield two groups of the same size? And why is age given such consideration, compared to other factors that are well known to affect vitamin D levels as outlined above? Furthermore, the authors should consider using analysis of covariance instead of t-tests, so they can include important covariates that may affect the results, as outlined above.
- Results. Figure 1 and Table 1 have some redundancy in terms of what they show, yet neither one shows p-values comparing the vitamin D concentrations between the age groups within each deployment category, or between those deployed to a war zone versus not. The authors should remove the redundancies and add p-values as necessary. Additional descriptive characteristics of the participants, such as percent female, percent belonging each ethnic group, body mass index and physical activity (both of which also affect vitamin D concentrations) would also be good to report.
- Table 2 does not need to show everything that it does. It looks like output from SPSS, reformatted and pasted into Word. All that is truly necessary is the “% within Deploy” for each vitamin D category, as well as the p-value for the chi-square test (and the counts, which could be part of the legend if this table were turned into a figure, which this reviewer recommends).
- Figure 3 should show percentages for each vitamin D category, rather than counts. It should also show any p-values obtained from chi-square testing.
- The Discussion states: “This case study also raises other questions, “Are symptomatic mobilizing and demobilizing U.S service members at higher risk of vitamin D deficiency and of requiring long term care for chronic multi-symptom illnesses through the VA Health Care System than asymptomatic mobilizing and demobilizing service members?” This is, indeed, an important question, but the authors must be very careful not to imply that their data suggest that those who are symptomatic have lower vitamin D levels, because they did not compare these data with those obtained from asymptomatic military personnel. In fact, if the authors were able to obtain vitamin D plasma samples from asymptomatic personnel at the same site, the comparison would make this manuscript much stronger. This is, in effect, the study that the authors propose in the following paragraph. If the comparison data were available, it would make more sense to comment on herpetic viruses and immunity, which the authors do both in the introduction and the discussion; in this reviewer’s opinion, these comments are perhaps too abundant given that the data presented really do not show anything related to either Epstein-Barr or immunity and inflammation.
Reviewer 2 Report
Thank you for inviting me to review this interesting paper. It is generally well written and is novel as there is a lack of research into this subject. I just have some minor comments.
Title- ‘Latent herpetic inflammatory response’ must be removed from the title as the paper doesn’t measure this at all (in terms of actual results) and it is misleading. Also, it might be better to say something like ‘Low Vitamin D States observed in US Marines and Navy Sailors with Early multi symptom illness’ as you don’t show that there is an association between the illness and vitamin D (because healthy soldiers/sailors could theoretically be deficient also- you didn’t include a healthy control group and demonstrate that these had sufficient vitamin D).
Abstract
Line 25 Can you rephrase? The data doesn't suggest they should be screened for symptoms of multi system illness- the data just show that those who have multi symptom illness (the participants in your study) should be checked for vitamin D.
Introduction
Line 42 Can you rephrase? This is not the normal treatment for all multi system illnesses (at least not for ME/CFS- I am not sure about gulf war type illness as this is not my knowledge base). Perhaps you could say that this type of therapy is experimental at present and not routine.
Line 44 I would say 25-hydroxyvitamin D (rather than vitamin D) to show that you are talking about the form that is already hydroxylated by the liver (not the precursor form). I would say activated rather than produced- sounds odd as it is produced in the skin.
Line 45 ‘uses’ would sound better than ‘has’
Methods
Line 116 Typo- ‘is’ should be ‘in’
Line 128 Bear in mind that these cut-offs are quite high- some European countries would consider 20 ng/mL (50nmol/L) as sufficient but your results are correct for the USA so okay to keep as it is.
Line 131 Typo- should be 'DiaSorin'
Results
Table 1: It would be useful to also have P values here- (as you cite them in the text). Otherwise it seems odd citing P values in the text but only 95% CI in the table.
Table 2: Should be % within deploy (not ‘within deploy')- missing % symbol
Figure 2 legend- can you add here the number of participants (n=6 in each group) juts to remind the reader that these are small samples.
Discussion
L235 It is odd why you only see age differences in the warzones and not other deployments. Perhaps this is a cohort effect whereby something was different about war zones in the past (or more likely the people deployed to them) compared with today. Does ethnicity differ in the older and younger cohorts? We know that Black people often have a lower 25(OH)D concentration compared with White people- if there are more Black personnel in the younger group then this could explain the difference in 25(OH)D by age. Could you mention ethnicity in the paper somewhere to either confirm this possibility or rule it out?
L247 Some other studies have found that 25(OH)D increases in soldiers during deployment to desert areas. For example research in males (not selected for multi-symptom illness) has found that 25(OH)D increases during deployment to Afghanistan. See https://www.ncbi.nlm.nih.gov/pubmed/30604661. It would be good to mention that not all studies find that 25(OH)D decreases during deployment and give an example where it increases.
Reviewer 3 Report
This paper studies medical records of US Marines or Sailors seeking medical attention, to see if deployment status or age is associated with vitamin D status. Though the study presents a unique possibility to investigate vitamin D status in military personnel, there aim of the study is unclear, the methods need clarification and the results should be expanded to show a more indepth analysis.
Major comments:
1) The aim of this study is unclear. The introduction section builds towards vitamin D status as a contributor to poor health, but the results only present vitamin D status in relation to age and deployment status.
2) The methods are unclear- especially how and when participants went through their medical check up and were assessed for vitamin D status. Some of the unclarity might be due to a lack of prior knowledge about US military procedures. It is unclear if:
- Participants were checked before, during or after deployment
- Participants were measured for 25OHD3 at the time, or using banked serum
- Participants sought medical attention for the symptoms described, or if these were present at routine check up
3) Seasonal correction of 25OHD may not be optimal for this study. Several factors might influence the seasonal contribution to 25OHD, and the disparate results from deployed and younger participants could be an artifact from the handling of 25OHD data. See comment 4 for a suggestion on how to handle 25OHD data.
4) Simple T-test and chi-square statistics are too basic for an observational study. These analyses should be supplemented with, or replaced by, analyses that allows for adjustment of confounding such as linear regression analysis/ANCOVA/mixed models. In such as model, season could be added as a covariate and thus adjusted for. Also, gender should be included.
5) Would it not be interesting to study 25OHD in relation to the different symptoms?
6) The title of the paper does not really represent its content. Consider revising it.
Round 2
Reviewer 1 Report
This version of the manuscript has improved significantly and the rationale is more clear. I still would like there to be a category for analysis of vitamin D deficiency as the Institute of Medicine, which sets the DRIs, defines it (<30 nmol/L), since this is the level where we know that true deficiency appears. The level chosen by the authors (<50 nmol/L) is generally considered insufficient, but it would be interesting to see what percentage of the military personnel in the study were below the most stringent definition of deficiency.
I understand that ethnicity, physical activity and other covariates were not available. I am glad that the authors have addressed this as a limitation. It is possible that more of the members who were deployed (and had lower vitamin D levels) were of non-European ancestry. That could have confounded the results, and this should be made more clear in that section.
In general, I do agree that these are interesting findings worthy of "getting out there" so that others will further evaluate the reasons behind the observed trends.
Author Response
1) “I would like there to be a category for analysis of vitamin D deficiency which sets vitamin D deficiency at < 30 nmole/L or 12.02 ng/ml based on the Institute of Medicine’s definition of deficiency.”
There were only three U.S. Marines in this study who had deficiency levels lower than 12.02 ng/ml. (i.e. 10, 9.4, and 11.4 ng/ml) There were no U.S. Navy Sailors who had serum 25(OH)D3 levels less than 12.02 ng/ml. There was one U.S. Marine and two U.S. Navy Sailors who had serum 25(OH)D3 levels in the range of 12 levels. (12.3, 12.8, and 12.9 ng/ml) From a clinical standpoint, the vast majority of clinicians in the U.S. Navy and at the VA, treat symptomatic patients with vitamin D levels less than 20 ng/ml with vitamin D3 supplementation.
The controversy concerning where to set normal ranges and deficiency levels for vitamin D is in part due to the fact that serum 25(OH)D3 levels do not always represent how much active vitamin D is actually being produced and the fact that many tissues of the body are acted upon by vitamin D not just bone for which deficiency, insufficiency, and normal ranges exist. We chose to use the ranges adapted by the reference labs which conducted our tests. Those ranges have been recommended by the Endocrine Society. This is in contrast to the above Institute of Medicine guideline.
The following sentence was added “The guideline for vitamin D deficiency recommended by the Endocrine Society is in contrast to the guideline for deficiency of 30 nm/L or 12.02 ng/ml recommended by the Institute of Medicine.” (lines 211-214)
2) “It is possible that more of the members who were deployed (and had lower vitamin D levels) of non-European ancestry. That could have confounded the results, and this could be made more clear.”
The following sentence was added “It is possible that more members who were deployed (and had lower vitamin D levels) were of non-European ancestry which could have confounded the results.” (lines 472-474)
Reviewer 3 Report
This paper has been improved greatly. However, some factors are still unclear and should be addressed before this manuscript is accepted for publication.
Major comments:
- The aim of the study is still unclear. The main aim was to review the medical charts, and see how many symptomatic (of what?) service members had poor vitamin D status. As the study sample are made up of service members with poor health, the vitamin D status of the group is unlikely to be representative of the underlying population. As no data on the symptoms related to 25OHD concentration are presented, as these were not retrieved, the relevance of the study aim can be questioned.
- The authors use quite controversial cut offs for 25OHD concentration without much discussion. The authors should consider adding the proportion below 12,5 or 15 mg/ml as well, as this is the most clinically relevant definition of vitamin D deficiency (ie for bone health, the only verified health outcome related to vitamin D status).
- A mean 25OHD concentration of 25-30 ng/ml is expected using the DIASORIN method, that tends to underestimate 25OHD compared to LC-MS/MS (the other method included). The authors state in their response that there was no difference in vitamin D status, depending on method. This is very surprising, and data to support these findings should be included. Please also include how many samples were analyzed using each method.
Minor comments:
- Could a table presenting the available characteristics of the study participants be included? It would be helpful to get an overview.
- Statistic methodology should be included in table footnotes.
Author Response
1) “The aim of the study is still unclear. “
The main aim of the study was to document low vitamin D status in symptomatic demobilizing veterans and other symptomatic cadre serving during the this very active time at Camp Lejeune. Currently, many veterans who have served during our ongoing conflicts in the middle east return with poorly defined but disabling conditions which the VA has currently defined as chronic multisymptom illnesses. These returning veterans were not known or recognized as being in poor health at the time of their evaluation based on known medical illnesses. Rather they had the kind of symptoms which we now see in many veterans which over time have become disabling for them and which the VA terms chronic multisymptom illnesses. The pathophysiology of this disorder is not fully known but there is growing evidence that the immune system and particularly herpetic latent viruses may play a role. Anecdotally correcting low vitamin D status has helped veterans with these symptoms and the modulation of the immune system by vitamin D may be playing a positive role in treating these patients.
2) “The Authors use quite controversial cut offs for 25(OH)D3 without much discussion.“
There were only three U.S. Marines in this study who had deficiency levels lower than 30 nmoles/L or 12.02 ng/ml. (i.e. 10, 9.4, and 11.4 ng/ml) There were no U.S. Navy Sailors who had serum 25(OH)D3 levels less than 12.02 ng/ml. There was one U.S. Marine and two U.S. Navy Sailors who had serum 25(OH)D3 levels in the range of 12 levels. (12.3, 12.8, and 12.9 ng/ml) From a clinical standpoint, the vast majority of clinicians in the U.S. Navy and at the VA, treat symptomatic patients with vitamin D levels less than 20 ng/ml with vitamin D3 supplementation.
The controversy concerning where to set normal ranges and deficiency levels for vitamin D is in part due to the fact that serum 25(OH)D3 levels do not always represent how much active vitamin D is actually being produced and the fact that many tissues of the body are acted upon by vitamin D not just bone for which deficiency, insufficiency, and normal ranges exist. We chose to use the ranges adapted by the reference labs which conducted our tests. Those ranges have been recommended by the Endocrine Society. This is in contrast to the Institute of Medicine recommended lower level for deficiency.
The following sentence was added “The guideline for vitamin D deficiency recommended by the Endocrine Society is in contrast to the guideline for deficiency of 30 nm/L or 12.02 ng/ml recommended by the Institute of Medicine.” (lines 211-214)
3) “A mean 25OHD concentration of 25-30 ng/ml is expected using the DIASORIN method, that tends to underestimate 25OHD compared to LC-MS/MS (the other method included). The authors state in their response that there was no difference in vitamin D status, depending on method. This is very surprising, and data to support these findings should be included. Please also include how many samples were analyzed using each method.”
During the period of this study September 2010 to July 2011, the number of serum 25(OH)D3 levels drawn from US. Service members both at the DPC and the Naval Hospital Camp Lejeune Sports Medicine Clinic (ordered by other medical providers) was increasing and was demonstrating a significant number of U.S. Marines with vitamin D deficiency which required treatment. In order to clarify whether this might be a reference lab problem, Naval Hospital Camp Lejeune changed their reference lab for vitamin D specimens from Lab Corp of America to Quest Labs. A comparison was made between the results from these two labs.
The following was added. One hundred and twenty-five samples were from Lab Corp and 71 from Quest Lab during the study period. Mean levels were 25.35 ng/ml from Lab Corps and 25.68 from Quest Lab. The lab that evaluated the data were not found to be related to the adjusted serum 25(OH)D3 levels, r = 0.078, p = 0.400. (Lines 204-207)
4) “Could a table presenting the available characteristics of the study participants be included?”
A table with the participant characteristics in each of the groups was added to Appendix A and a reference to the table is provided in the method section. (line 260-261) (lines 563-578)
5) Statistic methodology should be included in table footnotes.
The following was added.
Table 1 Other= Marines deployed to locations other than a war zone were not found to have age-related difference in serum levels, F = 0.65, p = 0.801. Wz= Marines deployed to the war zones of Iraq or Afghanistan were found to have an age-related difference in serum levels, F = 7.42, p = 0.008.
(lines 302-304)
Table 2. No difference in distribution by deployment groups (Chi Square(2,n=105)=. 0.35, p = 0.80). (lines 306-307)